# Loss Function Discovery for Object Detection via Convergence-Simulation Driven Search

**Peidong Liu**[1,*]**, Gengwei Zhang**[2,*]**, Bochao Wang**[3]**, Hang Xu**[3]
**Xiaodan Liang**[2,†]**, Yong Jiang**[1,†]**, Zhenguo Li**[3]

[1]Tsinghua Shenzhen International Graduate School, Tsinghua University
[2]Sun Yat-Sen University
[3]Huawei Noah's Ark Lab
lpd19@mails.tsinghua.edu.cn
{zgwdavid, sergeywong, chromexbjxh, xdliang328}@gmail.com
jiangy@sz.tsinghua.edu.cn    li.zhenguo@huawei.com

## Abstract

Designing proper loss functions for vision tasks has been a long-standing research direction to advance the capability of existing models. For object detection, the well-established classification and regression loss functions have been carefully designed by considering diverse learning challenges (e.g. class imbalance, hard negative samples, and scale variances). Inspired by the recent progress in network architecture search, it is interesting to explore the possibility of discovering new loss function formulations via directly searching the primitive operation combinations. So that the learned losses not only fit for diverse object detection challenges to alleviate huge human efforts, but also have better alignment with evaluation metric and good mathematical convergence property. Beyond the previous auto-loss works on face recognition and image classification, our work makes the first attempt to discover new loss functions for the challenging object detection from primitive operation levels and finds the searched losses are insightful. We propose an effective convergence-simulation driven evolutionary search algorithm, called CSE-Autoloss, for speeding up the search progress by regularizing the mathematical rationality of loss candidates via two progressive convergence simulation modules: convergence property verification and model optimization simulation. CSE-Autoloss involves the search space (i.e. 21 mathematical operators, 3 constant-type inputs, and 3 variable-type inputs) that cover a wide range of the possible variants of existing losses and discovers best-searched loss function combination within a short time (around 1.5 wall-clock days with 20x speedup in comparison to the vanilla evolutionary algorithm). We conduct extensive evaluations of loss function search on popular detectors and validate the good generalization capability of searched losses across diverse architectures and various datasets. Our experiments show that the best-discovered loss function combinations outperform default combinations (Cross-entropy/Focal loss for classification and L1 loss for regression) by 1.1% and 0.8% in terms of mAP for two-stage and one-stage detectors on COCO respectively. Our searched losses are available at https://github.com/PerdonLiu/CSE-Autoloss.

## 1 Introduction

The computer vision community has witnessed substantial progress in object detection in recent years. The advances for the architecture design, e.g. two-stage detectors (Ren et al., 2015; Cai & Vasconcelos, 2018) and one-stage detectors (Lin et al., 2017b; Tian et al., 2019), have remarkably

---

*Equal Contribution. Work done when the first author (Peidong Liu) interns at Huawei Noah's Ark Lab.
†Correspondence to: Xiaodan Liang (xdliang328@gmail.com), Yong Jiang (jiangy@sz.tsinghua.edu.cn).

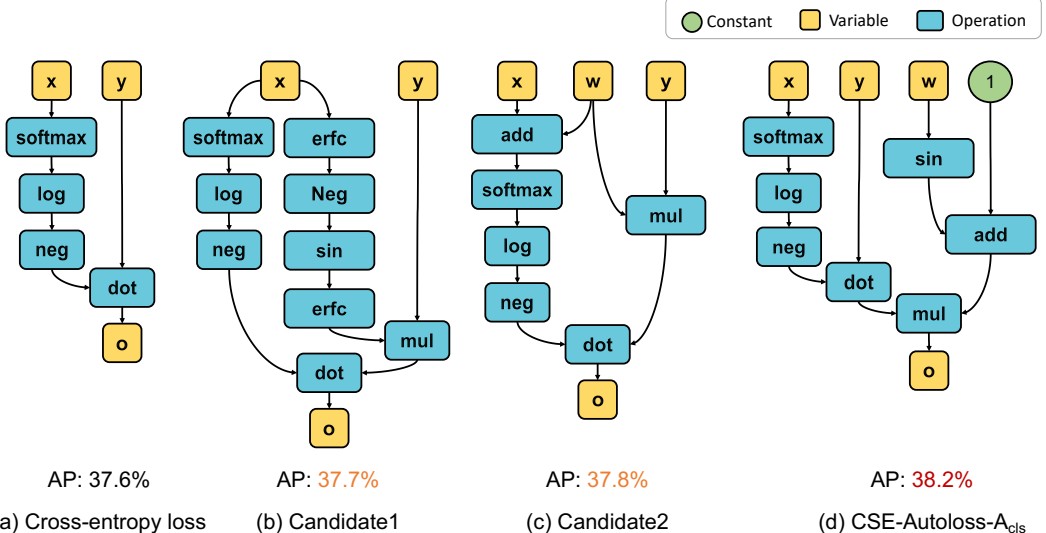

Figure 1: Computation graphs of loss function examples: (a) Cross-entropy loss; (b, c) Searched loss candidates with comparable performance to Cross-entropy loss; (d) Searched best-performed loss named CSE-Autoloss-A$_{cls}$.

pushed forward the state of the art. The success cannot be separated from the sophisticated design for training objective, i.e. loss function.

Traditionally, two-stage detectors equip the combination of Cross-entropy loss (CE) and L1 loss/Smooth L1 loss (Girshick, 2015) for bounding box classification and regression respectively. In contrast, one-stage detectors, suffering from the severe positive-negative sample imbalance due to dense sampling of possible object locations, introduce Focal loss (FL) (Lin et al., 2017b) to alleviate the imbalance issue. However, optimizing object detectors with traditional hand-crafted loss functions may lead to sub-optimal solutions due to the limited connection with the evaluation metric (e.g. AP). Therefore, IoU-Net (Jiang et al., 2018) proposes to jointly predict Intersection over Union (IoU) during training. IoU loss series, including IoU loss (Yu et al., 2016), Bounded IoU loss (Tychsen-Smith & Petersson, 2018), Generalized IoU loss (GIoU) (Rezatofighi et al., 2019), Distance IoU loss (DIoU), and Complete IoU loss (CIoU) (Zheng et al., 2020), optimize IoU between predicted and target directly. These works manifest the necessity of developing effective loss functions towards better alignment with evaluation metric for object detection, while they heavily rely on careful design and expertise experience.

In this work, we aim to discover novel loss functions for object detection automatically to reduce human burden, inspired by the recent progress in network architecture search (NAS) and automated machine learning (AutoML) (Cai et al., 2019; Liu et al., 2020a). Different from Wang et al. (2020) and Li et al. (2019b) that only search for particular hyper-parameters within the fixed loss formula, we steer towards finding new forms of the loss function. Notably, AutoML-Zero (Real et al., 2020) proposes a framework to construct ML algorithm from simple mathematical operations, which motivates us to design loss functions from primitive mathematical operations with evolutionary algorithm. However, it encounters a severe issue that a slight variation of operations would lead to a huge performance drop, which is attributed to the sparse action space. Therefore, we propose a novel Convergence-Simulation driven Evolutionary search algorithm, named CSE-Autoloss, to alleviate the sparsity issue. Benefit from the flexibility and effectiveness of the search space, as Figure 1 shows, CSE-Autoloss discovers distinct loss formulas with comparable performance with the Cross-entropy loss, such as (b) and (c) in the figure. Moreover, the best-searched loss function (d), named CSE-Autoloss-A$_{cls}$, outperformed CE loss by a large margin. Specifically, to get preferable loss functions, CSE-Autoloss contains a well-designed search space, including 20 primitive mathematical operations, 3 constant-type inputs, and 3 variable-type inputs, which can cover a wide range of existing popular hand-crafted loss functions. Besides, to tackle the sparsity issue, CSE-Autoloss puts forward progressive convergence-simulation modules, which verify the evolved loss functions

from two aspects, including mathematical convergence property and optimization behavior, facilitating the efficiency of the vanilla evolution algorithm for loss function search without compromising the accuracy of the best-discovered loss. For different types of detector, CSE-Autoloss is capable of designing appropriate loss functions automatically without onerous human works.

In summary, our main contributions are as follows: 1) We put forward CSE-Autoloss, an end-to-end pipeline, which makes the first study to search insightful loss functions towards aligning the evaluation metric for object detection. 2) A well-designed search space, consisting of various primitive operations and inputs, is proposed as the foundation of the search algorithm to explore novel loss functions. 3) To handle the inefficiency issue caused by the sparsity of action space, innovative convergence-simulation modules are raised, which significantly reduces the search overhead with promising discovered loss functions. 4) Extensive evaluation experiments on various detectors and different detection benchmarks including COCO, VOC2017, and BDD, demonstrate the effectiveness and generalization of both CSE-Autoloss and the discovered loss functions.

## 2 RELATED WORK

**Loss Functions in Object Detection.** In object detection frameworks, CE loss dominates the two-stage detectors (Ren et al., 2015; Cai & Vasconcelos, 2018) for classification purpose, while Focal loss (Lin et al., 2017b) and GHM loss (Li et al., 2019a) are widely equipped in one-stage detectors (Lin et al., 2017b; Tian et al., 2019) for solving imbalance issues between positive and negative samples. Chen et al. (2019b); Qian et al. (2020) attempt to handle the sample-imbalance problem from the ranking perspective. However, these works are sensitive to hyper-parameters hence they cannot generalize well. To further improve classification and localization quality, Jiang et al. (2018); Tian et al. (2019); Wu et al. (2020) introduce quality estimation, and GFocal (Li et al., 2020) unifies the quality estimation with classification towards consistent prediction. With regard to regression loss functions, Smooth L1 Loss (Girshick, 2015) has been commonly performed in the past years until IoU-based loss series (Yu et al., 2016; Rezatofighi et al., 2019; Zheng et al., 2020) gradually occupy regression branch for their effectiveness in bounding box distance representation due to the direct optimization of the evaluation metric. However, these works rely on expert experience on loss formula construction, which limits the development of the loss function design in object detection.

**Automated Loss Function Design.** By introducing a unified formulation of loss function, Li et al. (2019b); Wang et al. (2020) raise automated techniques to adjust loss functions for face recognition. However, they only search for specific hyper-parameters in the fixed loss formulas, which limit the search space and fail to discover new forms of loss functions. Real et al. (2020); Liu et al. (2020b); Ramachandran et al. (2018) attempt to design formulations from basic mathematical operations to construct building blocks in machine learning such as normalization layers and activation functions. However, these works cannot directly apply to loss function search since their search space and search strategies are specialized. Gonzalez & Miikkulainen (2020) proposes a framework for searching classification loss functions but the searched loss poorly generalizes to large-scale datasets. Besides, all these works are not suitable for the challenging object detection task due to the sparsity of the action space and the heavy evaluation cost for training object detectors. Instead, in this work, we make the first attempt to search for loss function formulations directly on the large-scale detection dataset via an effective pipeline with novel convergence-simulation modules.

## 3 METHOD

In this section, we present the CSE-Autoloss pipeline for discovering novel loss functions towards aligning the evaluation metric for object detection. We first introduce the well-designed search space in Section 3.1. The detailed CSE-Autoloss pipeline is elaborated in Section 3.2.

### 3.1 SEARCH SPACE DESIGN

**Input Nodes.** In object detection, default classification losses (i.e. CE loss and FL loss), take prediction and label as inputs. Inspired by GFocal loss (Li et al., 2020), to better motivate the loss to align with the evaluation metric (i.e. AP), we introduce IoU between ground truth and prediction into the loss formula, where prediction, label, IoU are notated as $x$, $y$, $w$ respectively. For the regression

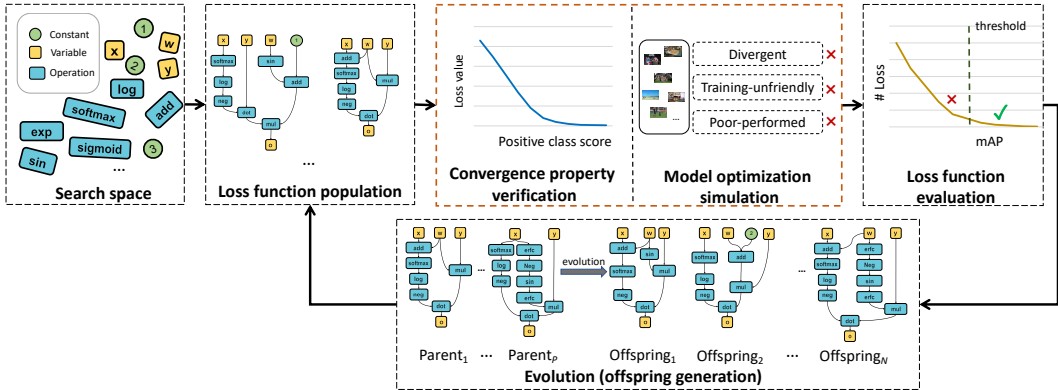

Figure 2: An overview of the proposed CSE-Autoloss pipeline. Our CSE-AutoLoss first generates a large amount of candidate loss functions by assembling formulations based on the well-designed search space. Then CSE-Autoloss filters out more than 99% divergent, training-unfriendly, or poor-performed loss candidates by evaluating them with the proposed convergence-simulation modules, i.e. convergence property verification and model optimization simulation, which verifies mathematical property and optimization quality. After that, $K$ top-performing loss functions are selected to evaluate on the proxy task to further obtain top-$P$ loss candidates as parents to derive offspring for the next generation. Please see more details about CSE-Autoloss in Algorithm 1.

branch, to cover the mainstream IoU loss series (i.e. IoU and GIoU), we take the intersection $i$, union $u$, and enclosing area $e$ between the predicted and target box as input tensors. Besides, 3 constant-type inputs (i.e. 1, 2, 3) are brought in to improve the flexibility of the search space.

As mentioned above, for consistent prediction of localization quality and classification score between training and testing, we simply introduce IoU into the CE and FL loss to get two nuanced loss variants, namely CEI and FLI respectively:

$$\text{CEI}(x, y, w) = -\text{dot}(wy, \log(\text{softmax}(x))),$$
$$\text{FLI}(x', y', w) = -w[y'(1 - \sigma(x')) \log \sigma(x') + (1 - y')\sigma(x') \log(1 - \sigma(x'))],$$

where $x, y \in \mathbb{R}^{c+1}$, $x' \in \mathbb{R}$, c is the number of class. $x$ and $x'$ are the prediction, $y$ is a one-hot vector of ground truth class, $y' \in \{0, 1\}$ is the binary target, and $w \in \mathbb{R}$ is the IoU value between ground truth and prediction. As Table 4 shows, CEI and FLI outperform CE and FL by a small margin in terms of AP, which verifies the effectiveness of introducing IoU into the classification branch. Note that we apply CEI and FLI loss as the predefined initial loss $\mathcal{L}_I$ in the search experiment for two-stage detectors and one-stage detectors, respectively.

**Primitive Operators.** The primitive operators, including element-wise and aggregation functions that enable interactions between tensors with different dimensions, are listed as below:

- Unary functions: $-x$, $e^x$, $\log(x)$, $|x|$, $\sqrt{x}$, softmax$(x)$, softplus$(x)$, $\sigma(x)$, gd$(x)$, alf$(x)$, erf$(x)$, erfc$(x)$, tanh$(x)$, relu$(x)$, sin$(x)$, cos$(x)$
- Binary functions: $x_1 + x_2$, $x_1 - x_2$, $x_1 \times x_2$, $\frac{x_1}{x_2 + \epsilon}$, dot$(x_1, x_2)$

$\epsilon$ is a small value to avoid zero division, softplus$(x) = \ln(1 + e^x)$, $\sigma(x) = \frac{1}{1 + e^{-x}}$ is the sigmoid function. To enlarge the search space, more S-type curves and their variants* are included, i.e. gd$(x) = 2\arctan(\tanh(\frac{x}{2}))$, alf$(x) = \frac{x}{\sqrt{1 + x^2}}$, erf$(x) = \frac{2}{\sqrt{\pi}} \int_0^x e^{-t^2} dt$ is the error function, erfc$(x) = 1 - \text{erf}(x)$. For binary operations, broadcasting is introduced to ensure the assembling of inputs with different dimensions. dot$(x_1, x_2)$ is the class-wise dot product of two matrix $x_1, x_2 \in R^{B \times C}$, where $B$ and $C$ indicate batch size and number of categories respectively. In practice, regression branch contains fewer primitive operators than classification branch because introducing more primitive functions for regression branch does not bring performance gain.

---

*In the implementation, the output of these functions are re-scaled to obtain the same range as sigmoid function, i.e. from 0 to 1.

**Loss Function as a Computation Graph.** As Figure 1 illustrates, we represent the loss function as a directed acyclic graph (DAG) computation graph that transforms input nodes into the scalar output $o$ (i.e. loss value) with multiple primitive operators in the intermediate.

**Sparsity of the Action Space.** We perform random search for Faster R-CNN classification branch on COCO and find only one acceptable loss every $10^5$, which indicates the sparsity of the action space and the challenge for searching the loss function formulations.

## 3.2 CSE-AUTOLOSS PIPELINE

We propose a novel CSE-Autoloss pipeline with progressive convergence-simulation modules, consisting of two perspectives: 1) convergence property verification module verifies the mathematical property of the loss candidates. 2) model optimization simulation module estimates the optimization quality to further filter out divergent, training-unfriendly, and poor-performed loss functions. An overview of CSE-Autoloss is illustrated in Figure 2 and the details are shown in Algorithm 1. Our CSE-Autoloss achieves more than 20x speedup in comparison to the vanilla evolutionary algorithm.

### 3.2.1 EVOLUTIONARY ALGORITHM

Here we give a short overview of the vanilla evolutionary algorithm, which refers to the traditional tournament selection (Goldberg & Deb, 1991). We first generate $N$ loss functions from a predefined initial loss $\mathcal{L}_I$ as the first generation, where a loss function indicates an individual in the population. Then these individuals perform cross-over randomly pair by pair with probability $p_1$ and conduct mutation independently with probability $p_2$. We evaluate all the individuals on the proxy task and select top-$P$ of them to serve as parents for the next generation. The above process is repeated for $E$ generations to get the best-searched loss.

The bottleneck of the evolutionary process lies at tons of loss evaluations on the proxy set due to the highly sparse action space, and it costs approximately half an hour for each evaluation. How to estimate the quality of individuals in the early stage is crucial for efficiency. Therefore, we apply a computational validness check for the population. Specifically, invalid values (i.e. NaN, $+\infty$, $-\infty$) are not allowed. This simple verification accelerates the search process by 10x. But a great number of divergent, training-unfriendly, and poor-performed loss functions still remain, which yet require a great amount of computation cost.

### 3.2.2 CONVERGENCE PROPERTY VERIFICATION

As stated above, the vanilla evolution search suffers from inefficiency issues due to the high sparsity of the action space and the unawareness of loss mathematical property. Therefore, we introduce the convergence property verification module into the evolutionary algorithm, which filters out either non-monotonic or non-convergent loss candidates within a short time.

**Classification Loss Function** We analyze the property of existing popular classification loss function (i.e. CE loss). For simplicity, we only consider binary classification scenario, which is known as binary Cross-entropy loss (BCE):

$$\text{BCE}(x) = -\ln(\frac{1}{1+e^{-x}}), \quad \frac{\partial \text{BCE}(x)}{\partial x} = -1 + \frac{1}{1+e^{-x}},$$

where $x$ indicates positive class score. Further analysis of BCE inspires us that binary classification loss should meet basic and general mathematical properties to guarantee validness and convergence in challenging object detection. We summarize these properties as follows:

- Monotonicity. The loss value should monotonically decrease w.r.t positive class score and increase w.r.t negative class score because large positive score or small negative score implies a well-performed classifier, which should have a small loss penalty.
- Convergence. When the positive class score tends to be $+\infty$, the loss gradient magnitude should converge to zero to ensure model convergence.

**Regression Loss Function** For the regression branch, consistency of loss value and distance between the predicted and target bounding box, named distance-loss consistency, should be confirmed. Distance-loss consistency claims that loss value is supposed to change consistently with distance, where loss value increases when the prediction moves away from the ground truth.

### 3.2.3 MODEL OPTIMIZATION SIMULATION

Although the convergence property verification module filters out either non-monotonic or non-convergent losses, it does not guarantee that the loss candidates are suitable for optimization. The sparse action space demands that there should be other techniques to ensure the quality of optimization. To address this issue, we propose the model optimization simulation module that inspects the optimization capability of the loss.

Specifically, we train and test the loss candidates on a small verification dataset $D_{verify}$, which is constructed by sampling only one image randomly from each category on benchmark dataset like COCO, to estimate the optimization quality. Then we apply AP performance on $D_{verify}$ of the top loss in the previous generation as a threshold to filter out divergent, training-unfriendly, and poor-performed loss candidates.

---

**Algorithm 1** CSE-Autoloss algorithm.

---

**Input:** Search Space $S$, Number of Generations $E$, Population Size $N$, Number of Parents $P$, Verification Set $D_{verify}$, Proxy Training Set $D_{train}$, Proxy Validation Set $D_{val}$, Initial Predefined Loss Function $\mathcal{L}_I$, Number of Loss to Evaluate on Verification Set $K$
 1: Evolving $N$ loss functions to generate loss function population from $\mathcal{L}_I$
 2: **for** $e \leftarrow 1, E$ **do**
 3:     Check loss function properties with convergence-simulation modules
 4:     Train and test on $D_{verify}$ and keep top-$K$ loss candidates
 5:     Evaluate on $D_{train}$ and $D_{val}$, and keep top-$P$ losses as parents
 6:     Generate $N$ offspring from the parents for the next generation
 7: **end for**
**Output:** Best-discovered loss function $\mathcal{L}_{best}$

---

## 4 EXPERIMENTS

**Datasets** We conduct loss search on large-scale object detection dataset COCO (Lin et al., 2014) and further evaluate the best-searched loss combinations on datasets with different distribution and domain, i.e. PASCAL VOC (VOC) (Everingham et al., 2015) and Berkeley Deep Drive (BDD) (Yu et al., 2020), to verify the generalization of the searched loss functions across datasets. **COCO** is a common dataset with 80 object categories for object detection, containing 118K images for training and 5K minival for validation. In the search experiment, we randomly sample 10k images from the training set for validation purpose. **VOC** contains 20 object categories. We use the union of VOC 2007 trainval and VOC 2012 trainval for training and VOC 2007 test for validation and report mAP using IoU at 0.5. **BDD** is an autonomous driving dataset with 10 object classes, in which 70k images are for training and 10k images are for validation.

**Experimental Setup** We apply Faster R-CNN and FCOS as the representative detectors for two-stage and one-stage, respectively, in the loss search experiments on COCO for object detection. We apply ResNet-50 (He et al., 2016) and Feature Pyramid Network (Lin et al., 2017a) as feature extractor. For FCOS, we employ common tricks such as normalization on bounding box, centerness on regression, and center sampling. Besides that, we replace centerness branch with the IoU scores as the target instead of the original design for FCOS and ATSS to better utilize the IoU information, which has slight AP improvement. Note that loss weights are set default as MMDetection (Chen et al., 2019a) but the regression weight for ATSS sets to 1 for the search loss combinations. To be consistent with the authors' implementation, we use 4 GPUs with 4 images/GPU and 8 GPUs with 2 images/GPU for FCOS and Faster-RCNN. Concerning the proxy task for loss function evaluation, we perform training on the whole COCO benchmark for only one epoch to trade off performance and efficiency. Our code for object detection and evolutionary algorithm are based on MMDetection (Chen et al., 2019a) and DEAP (Fortin et al., 2012).

In the search experiments, regression and classification loss functions are searched independently. For regression loss search, we take GIoU loss (Rezatofighi et al., 2019) as predefined initial loss $\mathcal{L}_I$ to generate the first loss function population, with CE and FL serving as the fixed classification loss for Faster R-CNN and FCOS, respectively. While for classification branch search, CEI and FLI loss are served as $\mathcal{L}_I$, with GIoU serving as the fixed regression loss.

## 4.1 RESULTS ON TWO-STAGE AND ONE-STAGE DETECTORS

We name the best loss combination searched with Faster R-CNN R50 (Ren et al., 2015) and FCOS R50 (Tian et al., 2019) as CSE-Autoloss-A and CSE-Autoloss-B respectively, with the subscript *cls* and *reg* indicating classification and regression branch. The searched formulations are as follows:

$$\text{CSE-Autoloss-A}_{\text{cls}}(x, y, w) = -\text{dot}((1 + \sin(w))y, \log(\text{softmax}(x))),$$

$$\text{CSE-Autoloss-A}_{\text{reg}}(i, u, e) = (1 - \frac{i}{u}) + (1 - \frac{i+2}{e}),$$

$$\text{CSE-Autoloss-B}_{\text{cls}}(x, y, w) = -[wy(1 + \text{erf}(\sigma(1 - y)))\log\sigma(x) + (\text{gd}(x) - wy)(\sigma(x) - wy)\log(1 - \sigma(x))],$$

$$\text{CSE-Autoloss-B}_{\text{reg}}(i, u, e) = \frac{3eu + 12e + 3i + 3u + 18}{-3eu + iu + u^2 - 15e + 5i + 5u}.$$

Table 1: Detection results on the searched loss combination and default loss combination for multiple popular two-stage and one-stage detectors on COCO val. Column **Loss** indicates loss for classification and regression branch in sequence.

| Detector | Loss | AP (%) | AP$_{50}$(%) | AP$_{75}$(%) | AP$_S$(%) | AP$_M$(%) | AP$_L$(%) |
|---|---|---|---|---|---|---|---|
| Faster R-CNN R50 | CE + L1 | 37.4 | 58.1 | 40.4 | 21.2 | 41.0 | 48.1 |
| | CE + GIoU | 37.6$^{+0.2}$ | 58.2$^{+0.1}$ | 41.0$^{+0.6}$ | 21.5$^{+0.3}$ | 41.1$^{+0.1}$ | 48.9$^{+0.8}$ |
| | CSE-Autoloss-A | **38.5$^{+1.1}$** | 58.6$^{+0.5}$ | 41.8$^{+1.4}$ | 22.0$^{+0.8}$ | 42.2$^{+1.2}$ | 50.2$^{+2.1}$ |
| Faster R-CNN R101 | CE + L1 | 39.4 | 60.1 | 43.1 | 22.4 | 43.7 | 51.1 |
| | CE + GIoU | 39.6$^{+0.2}$ | 59.2$^{-0.9}$ | 42.9$^{-0.2}$ | 22.6$^{+0.2}$ | 43.5$^{-0.2}$ | 51.5$^{+0.4}$ |
| | CSE-Autoloss-A | **40.2$^{+0.8}$** | 60.1$^{+0.0}$ | 43.7$^{+0.6}$ | 22.6$^{+0.2}$ | 44.3$^{+0.6}$ | 52.7$^{+1.6}$ |
| Cascade R-CNN R50 | CE + Smooth L1 | 40.3 | 58.6 | 44.0 | 22.5 | 43.8 | 52.9 |
| | CE + GIoU | 40.2$^{-0.1}$ | 58.0$^{-0.6}$ | 43.6$^{-0.4}$ | 22.4$^{-0.1}$ | 43.6$^{-0.2}$ | 52.6$^{-0.3}$ |
| | CSE-Autoloss-A | **40.5$^{+0.2}$** | 58.8$^{+0.2}$ | 44.1$^{+0.1}$ | 22.8$^{+0.3}$ | 43.9$^{+0.1}$ | 53.3$^{+0.4}$ |
| Mask R-CNN R50 | CE + Smooth L1 | 38.2 | 58.8 | 41.4 | 21.9 | 40.9 | 49.5 |
| | CE + GIoU | 38.5$^{+0.3}$ | 58.8$^{+0.0}$ | 41.8$^{+0.4}$ | 21.9$^{+0.0}$ | 42.1$^{+1.2}$ | 49.7$^{+0.2}$ |
| | CSE-Autoloss-A | **39.1$^{+0.9}$** | 59.3$^{+0.5}$ | 42.4$^{+1.0}$ | 22.4$^{+0.5}$ | 43.0$^{+2.1}$ | 51.4$^{+1.9}$ |
| FCOS R50 | FL + GIoU | 38.8 | 56.8 | 42.2 | 22.4 | 42.6 | 51.1 |
| | CSE-Autoloss-B | **39.6$^{+0.8}$** | 57.5$^{+0.7}$ | 43.1$^{+0.9}$ | 22.7$^{+0.3}$ | 43.7$^{+1.1}$ | 52.6$^{+1.5}$ |
| ATSS R50 | FL + GIoU | 40.0 | 57.9 | 43.3 | 23.8 | 43.7 | 51.3 |
| | CSE-Autoloss-B | **40.5$^{+0.5}$** | 58.3$^{+0.4}$ | 43.9$^{+0.6}$ | 23.3$^{-0.5}$ | 44.3$^{+0.6}$ | 52.5$^{+1.2}$ |

The quantitative results about the gain brought by the searched loss under the same hyper-parameters for multiple popular two-stage and one-stage detectors are shown in Table 1. Results on Faster R-CNN R50 and FCOS R50 indicate the generalization of CSE-Autoloss across detectors and the best-searched loss combination is capable of stimulating the potential of detectors by a large margin. Take Faster R-CNN R50 for example, CSE-Autoloss-A outperforms the baseline by 1.1% in terms of AP, which is a great improvement for that two years of expert effort into designing hand-crafted loss function yields only 0.2% AP improvement from Bounded IoU loss to GIoU loss.

To verify the searched loss can generalize to different detectors, we respectively apply CSE-Autoloss-A on other two-stage models (i.e. Faster R-CNN R101, Cascade R-CNN R50 (Cai & Vasconcelos, 2018), and Mask R-CNN R50 (He et al., 2017)), and CSE-Autoloss-B on ATSS (Zhang et al., 2020). Note that we only replace the classification and regression branch of Mask R-CNN with CSE-Autoloss-A. Results in Table 1 represent the consistent gain brought by the effective searched loss without additional overhead.

We further conduct best-searched loss evaluation experiments for Faster R-CNN R50 on VOC and BDD to validate the loss transferability across datasets. Results are displayed in Table 3, which indicate the best-searched loss combination enables the detectors to converge well on datasets with different object distributions and domains.

Table 2: Efficiency improvement with progressive convergence-simulation modules on searching for Faster R-CNN classification branch. The proposed modules filter out 99% loss candidates, which enhances the efficiency to a large extent.

| Convergence property verification | Model optimization simulation | #Evaluated loss |
|:---:|:---:|:---:|
| | | $5 \times 10^3$ |
| $\checkmark$ | | $7 \times 10^2$ |
| $\checkmark$ | $\checkmark$ | 50 |

Table 3: Detection results on searched loss combination and default loss combination with Faster R-CNN on PASCAL VOC 2007 test and BDD val.

| Loss | VOC mAP (%) | BDD AP (%) |
|:---:|:---:|:---:|
| CE + L1 | 79.5 | 36.5 |
| CE + GIoU | $79.6^{+0.1}$ | $36.6^{+0.1}$ |
| CSE-Autoloss-A | $\mathbf{80.4^{+0.9}}$ | $\mathbf{37.3^{+0.8}}$ |

Table 4: Detection results on predefined initial loss and default loss for Faster R-CNN and FCOS classification branch on COCO val.

| Detector | Loss | AP (%) |
|:---:|:---:|:---:|
| Faster R-CNN R50 | CE + GIoU | 37.6 |
| | CEI + GIoU | $\mathbf{37.7^{+0.1}}$ |
| FCOS R50 | FL + GIoU | 38.8 |
| | FLI + GIoU | $\mathbf{39.0^{+0.2}}$ |

Table 5: Comparison on the efficiency of search algorithms and distinct branches for Faster R-CNN.

| Search algorithm | Search branch | Another branch | #Evaluated loss | Wall-clock hours | AP (%) |
|:---:|:---:|:---:|:---:|:---:|:---:|
| Random search | Classification | GIoU | $1 \times 10^4$ | $8 \times 10^3$ | 37.8 |
| Evolution search | Classification | GIoU | $5 \times 10^3$ | $5 \times 10^2$ | 38.2 |
| CSE-Autoloss | Classification | GIoU | 50 | 26 | 38.2 |
| CSE-Autoloss | Regression | CE | 15 | 8 | 37.9 |

## 4.2 ABLATION STUDY

**Effectiveness of Convergence-Simulation Modules.** Efficiency improvement with progressive convergence-simulation modules is shown in Table 2 on searching for Faster R-CNN classification branch. With our proposed modules, CSE-Autoloss filters out 99% loss candidates with bad mathematical property and poor performance, which largely increases the efficiency without compromising the accuracy of the best-discovered loss.

**Individual Loss Contribution.** As shown in Table 6 and Table 7, our searched losses match or outperform the existing popular losses consistently for Faster R-CNN and FCOS. Figure 3 illustrates the convergence behaviors of popular loss combinations and our CSE-Autoloss-A.

Table 6: Comparison on different loss combinations for Faster R-CNN R50 on COCO val.

| Loss | AP (%) |
|:---:|:---:|
| CE + L1 | 37.4 |
| CE + IoU | $37.9^{+0.5}$ |
| CE + Bounded IoU | $37.4^{+0.0}$ |
| CE + GIoU | $37.6^{+0.2}$ |
| CE + DIoU | $37.9^{+0.5}$ |
| CE + CIoU | $37.8^{+0.4}$ |
| CE + CSE-Autoloss-A$_{\mathrm{reg}}$ | $37.9^{+0.5}$ |
| CSE-Autoloss-A$_{\mathrm{cls}}$ + GIoU | $38.2^{+0.8}$ |
| CSE-Autoloss-A | $\mathbf{38.5^{+1.1}}$ |

Table 7: Comparison on different loss combinations for FCOS R50 on COCO val.

| Loss | AP (%) |
|:---:|:---:|
| FL + GIoU | 38.8 |
| FL + DIoU | $38.7^{-0.1}$ |
| FL + CIoU | $38.8^{+0.0}$ |
| GHM (Li et al., 2019a) | $38.6^{-0.2}$ |
| FL + CSE-Autoloss-B$_{\mathrm{reg}}$ | $39.1^{+0.3}$ |
| CSE-Autoloss-B$_{\mathrm{cls}}$ + GIoU | $39.4^{+0.6}$ |
| CSE-Autoloss-B | $\mathbf{39.6^{+0.8}}$ |

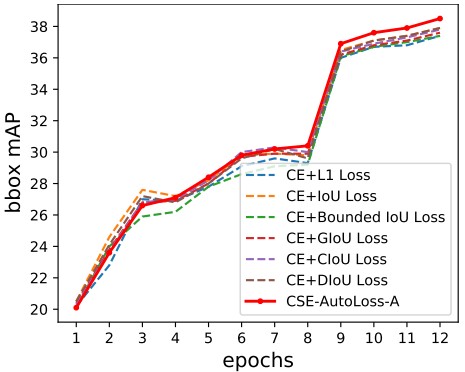

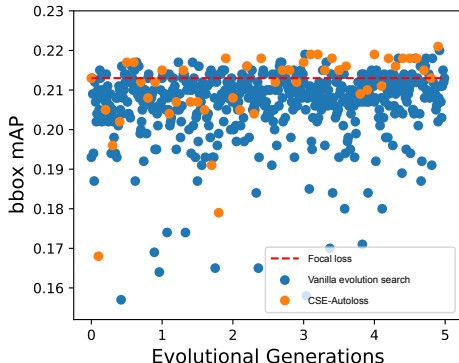

Figure 3: Evaluation results of popular hand-crafted loss combinations and CSE-Autoloss-A with Faster R-CNN R-50 on COCO val. Different loss functions have various convergence behaviors, and CSE-Autoloss-A achieves the best performance compared with other loss combinations by a large margin.

Figure 4: Visualization of the CSE-Autoloss and vanilla evolution search for FCOS, where the evaluated loss individuals are scattered as orange and blue respectively. With our convergence-simulation modules, the number of evaluated loss candidates is largely reduced.

**Search Algorithm.** We compare the efficiency between different search algorithms including random search, vanilla evolution search, and CSE-Autoloss. Table 5 shows that the evolution search is much better than random search. But due to the high sparsity of the action space, the vanilla evolution search requires hundreds of wall-clock hours for a server with 8 GPUs. CSE-Autoloss speeds up the search process by 20x because of the effective convergence-simulation modules and discovers well-performed loss combinations in around 1.5 wall-clock days.

**Search Complexity of Different Branches.** In our search space, the input nodes of the regression branch are scalars, while the classification branch contains vectors. Besides, the optimization of classification branch is more sensitive and difficult than regression, leading to sparse valid individuals. Therefore, the evolution population required for classification is $10^4$, while only $10^3$ for regression. The total number of evaluated loss and search overhead for discovering classification branch is much larger than that for regression as Table 5 shows.

**Evolution Behavior Analysis.** To validate the effectiveness of CSE-Autoloss, we further compare the behaviors of CSE-Autoloss and vanilla evolution search with FCOS. The vanilla evolution search evaluates around 1000 loss candidates in each generation, compared to only about 10 individuals for CSE-Autoloss. The loss individuals are scattered in Figure 4. Note that we only plot top-100 well-performed losses in each generation in the vanilla evolutionary search for clear visualization. The vanilla evolution search is hard to converge because a discrete replacement of the operation could fail a loss function. With our convergence-simulation modules, the efficiency is largely improved.

## 5 CONCLUSION

In this work, we propose a convergence-simulation driven evolutionary search pipeline, named CSE-Autoloss, for loss function search on object detection. It speeds up the search process largely by regularizing the mathematical property and optimization quality and discovers well-performed loss functions efficiently without compromising the accuracy. We conduct search experiments on both two-stage detectors and one-stage detectors and further empirically validate the best-searched loss functions on different architectures across datasets, which shows the effectiveness and generalization of both CSE-Autoloss and the best-discovered loss functions. We hope our work could provide insights for researchers of this field to design novel loss functions for object detection and more effective frameworks for automated loss function search.

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
