# OpenReview forum: "Loss Function Discovery for Object Detection via Convergence-Simulation Driven Search"
_ICLR.cc/2021/Conference — ICLR 2021 Poster_

### Official Review · AnonReviewer2 · 2020-10-28
**Evolutionary algorithm for searching loss function for object detectors**

**Rating:** 7
**Confidence:** 5

**Review:**

This paper proposes to use an evolutionary search algorithm to search for better loss functions for the classification and regression branch of an object detector. The algorithm starts with 20 primitive mathematical operations. Due to the highly sparse action space, the vanilla evolutionary algorithm would take a long time to converge. Then the authors propose two ways to reduce the search space. First, they filter out loss functions which generates gradients of large magnitude to well-classified samples and do not converge to zero. Second, they construct a very small dataset by sampling only one image randomly from each category and evaluate the loss function on it to quickly filter out bad loss candidates.

Pros:
The idea of applying evolutionary algorithm to search for a loss function for training an object detector is interesting and, to the best of my knowledge, new. The proposed way to reduce the search space seems to be effective. The authors apply the new loss function to other detectors, and it seems to generalize well to different detectors.

Cons:
The baselines provided in this paper are little bit weak. They are pretty far behind from the current state-of-the-art detectors. Although the new loss function is able to improve the performance of all of the baselines, it is unclear if it would generalize well to state-of-the-art results. The authors should consider using baselines with better performance. I also wonder if the loss would generalize to different datasets, such as PASCAL VOC.

In a subsection of section 3.1, “Effectiveness of the Search Space”, the authors mention that they introduce IoU into the cross entropy and focal loss, and show that they outperform the conventional cross entropy and focal loss. They then said this verifies the effectiveness of the search space. I am not seeing how this is connected to the effectiveness of the search space. Can the authors further elaborate on this?

The authors use the modified version of the cross entropy or focal loss as the initial loss in the search experiments. The final version of the loss is still pretty much the same as the initial loss. It seems that the search is biased towards to the initial loss. I wonder if the authors have tried other initializations.

The authors said that the regression branch contains fewer primitive operators because of the simplicity of the regression task. It is unclear which task, classification or regression, is actually simpler. It seems that regression may actually be harder as we have achieved human level results on the task of image classification but not on the detection, which requires precise localization of objects.

---

### Official Review · AnonReviewer4 · 2020-10-28
**Official Blind Review #4**

**Rating:** 6
**Confidence:** 4

**Review:**

Summary:

In this paper, the authors propose CSE-Autoloss to search loss functions for object detection. A well-designed search space is carefully designed to explore novel loss functions. In order to reduce the search overhead, convergence-simulation modules are raised.

Reasons for score:

Overall, I vote for accepting. My major concern is about the clarity of the paper and some additional experiments (see cons below). Hopefully the authors can address my concern in the rebuttal period.

Pros:

1.This paper proposes a new framework to search loss functions for object detection. For me, it is the first attempt to discover loss functions for object detection automatically.

2.To search the novel loss functions for object detection, the authors propose a well-designed search space.

3.To reduce the search overhead caused by evolutionary algorithm, this paper raises two modules: convergence property verification module and model optimization simulation module.

Cons:

1.For the predefined initial loss, CEI/FLI loss and GioU loss are used for classification and regression, respectively. If using other losses as the predefined initial loss, how is the performance? Could you provide some experiments?

2.In the search experiments, regression and classification loss functions are searched independently. For me, this can not result in the optimal loss functions. Why does not search jointly?

3.In the evolutionary algorithm, how do the probabilities of cross-over and mutation affect the performance of searched loss?

4.In the experiments, there are some results to verify the generalization of the searched loss on different detectors and datasets. If directly search loss functions on these detectors and datasets, how are the performances?

Questions during rebuttal period:

Please address and clarify the cons above

---

### Official Review · AnonReviewer1 · 2020-10-29
**An effective convergence-simulation driven evolutionary algorithm for searching loss function for object detection**

**Rating:** 6
**Confidence:** 4

**Review:**

This paper proposes CSE-Autoloss to deal with searching loss function for object detection with an effective convergence-simulation driven evolutionary search algorithm. CSE-Autoloss outperforms both previous CE/FL loss for classification and L1 loss for regression.

Pros:
1. Motivation is clear and good.
Loss design is tricky in object detection, since for it needs to deal with class imbalance and fore/background imbalance for classification loss and regression loss. This paper motivated by network architecture search, proposes effective searching algorithm for loss function, which is impressive and novel.

2. Convergence property verification and model optimization simulation are simple and effective.
Previous NAS papers usually require large computation resources and time to search the best model. CSE-Autoloss employs monotonicity and convergence of classification loss to and a small verity dataset to easily filter out loss candidate, it speedups the searching process by 10x and reaches similar results with random research in Table 5.

Cons:
1. Some related works on improving Focal loss and GIoU [1][2] are missing, it is better to include them as parts of hand-crafted loss functions for comparisons.

2.Is it possible to search classification and regression loss together? It seems for now classification and regression loss are searched independently.

3. Different formulations of CSE-Autoloss-A_cls and CSE-Autoloss-B_cls.
I understand that formulation of Autoloss_A_reg can be different from Autoloss_B_reg since their hand-crafted counterpart L1 & IoU loss are different, but for classification loss, FL loss is CE loss with a simple modulating factor, but it seems Autoloss-A_cls and Autoloss-B_cls design are quiet different. Any possible explanation for this difference?

[1] Distance-IoU Loss: Faster and Better Learning for Bounding Box Regression, in AAAI 2020
[2] Gradient Harmonized Single-stage Detector, in AAAI 2019

---

### Official Review · AnonReviewer3 · 2020-11-02
**Incremental improvements based on previous auto loss for image classification**

**Rating:** 5
**Confidence:** 4

**Review:**

This paper proposes to automatically discover proper loss functions for object detection. It first designs some unit mathematical operations as search space, and then performs evolutionary algorithm to discover well-performed loss functions for the object detection tasks. Different from image classification, one needs to search both classification and localization losses in object detection. To accelerate the search, the paper proposes convergence property verification and model optimization simulation to effectively evaluate the searched loss and reduce the search space.

Overall, this paper brings new ideas and experimental conclusions to the auto loss discovery in the object detection area. However, the proposed method is heavily based on the works of auto loss for image classification, i.e. "Improved Training Speed, Accuracy, and Data Utilization Through Loss Function Optimization". Some key differences lies on the acceleration by checking the Monotonicity and Convergence of the searched loss. Another issue I am curious about is how the method balances between classification and localization losses. Independently searching either cls or localization loss does not guarantee a good performance when combining them together for a high AP. Finally, the searched loss formulation seems complex and tricky, thus raising the doubt of its generalization across a wide range of neural networks and datasets. Do we need to re-search the loss function once we change the network architecture or the training data slightly?

---

### Decision · Program_Chairs · 2021-01-07
**Final Decision**

**Decision:**

Accept (Poster)

**Comment:**

This paper received borderline scores but overall lean positive.

The reviewers point out that the paper presents interesting new ideas and an effective solution to the problem of automatically searching for loss functions. The empirical results are convincing, although the baselines are not the strongest possible in terms of absolute performance. Overall, the ACs find that the paper has sufficient novelty and technical contribution to be accepted.